# Next-Generation Technologies and Strategies for the Management of Retinoblastoma

**DOI:** 10.3390/genes10121032

**Published:** 2019-12-11

**Authors:** Harini V. Gudiseva, Jesse L. Berry, Ashley Polski, Santa J. Tummina, Joan M. O’Brien

**Affiliations:** 1Scheie Eye Institute, University of Pennsylvania, Philadelphia, PA 19104, USA; gudiseva@pennmedicine.upenn.edu; 2The Vision Center at Children’s Hospital Los Angeles, Los Angeles, CA 90027, USA; Jesse.Berry@med.usc.edu (J.L.B.); ashley.polski@usc.edu (A.P.); 3Keck School of Medicine, University of Southern California, Los Angeles, CA 90033, USA; 4Office of the Director, National Eye Institute, National Institutes of Health, Bethesda, MD 20892, USA; tumminias@nei.nih.gov

**Keywords:** retinoblastoma, eyeGENE^®^, diagnostic testing, chemotherapy

## Abstract

Retinoblastoma (RB) is an inherited retinal disorder (IRD) caused by the mutation in the *RB1* gene or, rarely, by alterations in the *MYCN* gene. In recent years, new treatment advances have increased ocular and visual preservation in the developed world. The management of RB has improved significantly in recent decades, from the use of external beam radiation to recently, more localized treatments. Determining the underlying genetic cause of RB is critical for timely management decisions. The advent of next-generation sequencing technologies have assisted in understanding the molecular pathology of RB. Liquid biopsy of the aqueous humor has also had significant potential implications for tumor management. Currently, patients’ genotypic information, along with RB phenotypic presentation, are considered carefully when making treatment decisions aimed at globe preservation. Advances in molecular testing that improve our understanding of the molecular pathology of RB, together with multiple directed treatment options, are critical for developing precision medicine strategies to treat this disease.

## 1. Introduction

Inherited retinal diseases (IRDs) encompass a variety of vision-threatening ocular disorders that are caused by familial gene mutations and often follow Mendelian inheritance [1]. In order to develop precise therapeutics for IRDs, it is critical to identify the specific genetic defects that cause them. However, genetic diversity and multiple overlapping retinal phenotypes complicate our genetic and molecular understanding of these retinal dystrophies.

In recent decades, novel genotyping technologies—including positional cloning, linkage analysis, homozygosity mapping, candidate gene analyses, and whole exome sequencing (WES)—have improved our knowledge of the molecular underpinnings of IRDs. Mutations in greater than 300 genes and mapped loci are now known to be associated with specific retinal disorders (RetNet; https://sph.uth.edu/retnet/) [2]. Whole exome and next generation sequencing studies are currently on the rise to identify additional IRD-associated genes across different populations [3,4,5,6,7]. 

Retinoblastoma (RB), a pediatric malignancy of the developing retina, is a prototypical genetic cancer and IRD [8]. Usually diagnosed in children under the age of three [9], RB comprises an important subject in cancer genetics research due to its distinct association with the *RB1* tumor suppressor gene. While the majority of RB cases are caused by sporadic somatic mutations of the gene, upwards of 40% of patients are affected by a heritable form of the disease that usually affects both eyes and can lead to a host of other systemic malignancies [8,10,11,12,13]. In order to decipher the specific subtype (heritable versus non-heritable) of RB in presenting patients and their families, genetic testing of the peripheral blood to determine the presence of a germline mutation is an essential component in the management and counseling of these patients [14].

## 2. Retinoblastoma

RB is an ocular malignancy that forms in the developing retina [8]. It presents during childhood and infancy and is caused by a mutation in *RB1—*the first identified tumor suppressor gene [15,16,17]. The most common presenting sign of retinoblastoma is leukocoria, in which the normal red pupillary reflex is replaced with a whitish discoloration secondary to abnormal growth and calcification of the growing intraocular tumor [9]. Children may also present with strabismus, or misalignment of the eyes, due to loss of central vision from the growing tumor. Such presenting signs are common to both heritable and non-heritable forms of the disease, and their timely identification is important to achieve early diagnosis and treatment initiation. In developing countries, lack of available healthcare can lead to a significant delay in diagnosis, and children may present with extraocular disease. This not only reduces the rates of vision retention and ocular preservation but significantly increases the morbidity and mortality of disease for those children [18].

The incidence of retinoblastoma is approximately 1/15–20000 live births [19,20]. While no gender or ethnic disparities have been identified, a disproportionately greater number of cases are observed in populations with high birthrates [20,21]. Outcomes of retinoblastoma treatment also vary with geographical location. In certain Asian and African countries, for example, approximately 40–70% of children diagnosed with retinoblastoma die every year, compared to only 3–5% mortality rate in Europe, Canada and the United States (US) [22]. In the US, an estimated 300 children are diagnosed annually, with three out of four children demonstrating unilateral involvement. Although the majority of unilateral RB cases are caused by isolated, somatic genetic defects (in the tumor only), up to 15% of children with unilateral RB have an underlying germline mutation in the *RB1* gene (similar to patients with bilateral disease) [13,23]. While there are no known associations with the specific pathogenic *RB1* mutation and ocular outcomes (e.g., the ability to salvage the eye and retain vision), treatment decisions, prognosticating the risk of secondary tumors, and family counseling are critically affected by the knowledge of whether this disease is inherited or somatic.

## 3. Retinoblastoma Therapy

Retinoblastoma management has improved drastically over the past decades, with a heightened emphasis on ocular and vision preservation. It should be emphasized, however, that despite improvements in our ability to save the eye, these ocular conservation methods should only be undertaken if there are no high-risk clinical features that suggest pursuing this course of treatment puts the child at risk of metastatic disease. These high-risk clinical features include eyes that are functionally and physically destroyed by tumor, have no view of the tumor or other ocular structures due to vitreous hemorrhage or massive tumor necrosis, or have neovascular glaucoma [24,25,26]. These eyes generally respond poorly to treatment, and enucleation is a reasonable and safe primary therapy. Additionally, any suggestion of optic nerve enlargement or enhancement on MRI scanning mediates enucleation with attention to resecting the fullest possible extent. Conservative therapy for retinoblastoma in an attempt to save the eye(s) is most often done with chemotherapy, either as 3-drug intravenous or single/multi-agent intra-arterial chemotherapy [12,27,28,29]. Either regimen often requires consolidation with other localized therapies including red or green laser therapy, cryotherapy, intravitreal chemotherapy for tumor seeding in the vitreous, or brachytherapy for localized tumors. External beam radiation therapy, previously a common standard of care for these patients, is now used only in very rare instances, usually when tumor has recurred in the last remaining eye. Currently, treatment decisions are made based on clinical features only; a biopsy is never indicated for diagnosis of retinoblastoma. Furthermore, while the presence of a germline *RB1* remains an important factor in deciding appropriate follow-up strategies for patients with RB, the status is almost never known at diagnosis. Children with bilateral disease are of course presumed to have a germline mutation, and those with unilateral disease undergo careful screening of the normal eye with examinations under anesthesia, as well as optical coherence tomography imaging of the retina [30,31] until the *RB1* mutation status in the peripheral blood is confirmed. Thus, genetic testing remains an important component of the RB patient work-up.

## 4. Molecular Testing in Retinoblastoma

In order to screen [32] individuals for their susceptibility to RB and to better understand the nature and inheritance of RB, a spectrum of molecular testing approaches have been developed in recent years to identify pathogenic variants in the *RB1* gene [33,34,35]. In a Clinical Laboratory Improvement Amendments (CLIA)-regulated lab, DNA from peripheral blood is tested for single genes and also for chromosomal abnormalities by using genomic hybridization techniques such as chromosomal microarray analyses (CMA). Mutation identification is performed either by traditional Sanger sequencing of the amplified target regions [35,36] or by more recent technologies such as SNP arrays and next generation sequencing (NGS) strategies. 

Great strides have been made in improving the accuracy and efficiency of genetic testing for RB. This is particularly important because tumor tissue for the analysis of somatic mutations is only available if the eye has been enucleated (i.e., removed); therefore, the accuracy of the peripheral blood test needs to be high in the absence of the tumor mutation(s). NGS and *RB1* custom array-comparative genomic hybridization (aCGH) methods were recently used on a cohort of RB patients to optimize diagnostic procedures and to identify genomic abnormalities in retinoblastoma [37]. This combined, cost-effective approach was able to accurately detect point mutations, macrodeletions, and duplications from exon 18 to exon 23, and it could detect the level of mosaicism down to 1% [37]. Targeted NGS using Illumina MiSeq and precision bioinformatic pipelines were also used to identify a spectrum of pathogenic variants in retinoblastoma patients [38]. Using these comprehensive approaches, an array of novel pathogenic variants—including single nucleotide variants, InDels (insertions/deletions), and copy number variations—were detected in RB patients [37,38] and the time and number of assays required for detection of *RB1* pathogenic variants were reduced. Another recent method using a combination of direct sequencing and multiplex ligation-dependent probe amplification (MLPA) has also enhanced the detection of gross mutations and is able to identify germline mosaicism and genomic abnormalities spanning the promoter, exons and introns in *RB1* [39]. Germline parental mosaicism can also be detected with more sensitive methods like allele-specific PCR or NGS [40,41].

## 5. Retinoblastoma Genetics and Genomics

The variety of technologies in molecular genetics has facilitated more detailed characterizations of RB genetics and inheritance. The *RB1* gene is located on chromosome 13q14.1–q14.2, spans 180 kb and consists of 27 exons; it encodes an mRNA transcript of 4.7 kb and a protein of nearly 106 kDa. The unphosphorylated RB1 protein is involved in cell cycle regulation by binding to E2F transcription factor 1, ultimately resulting in cell cycle arrest [42]. Therefore, when this regulatory function is lost, such as in the setting of retinoblastoma, abnormal progression through the cell cycle can lead to cell proliferation and tumor development.

Biallelic inactivation of the RB1 gene is required to initiate RB development either by mutations in both RB1 alleles or more commonly due to loss of heterozygosity. Additional genetic and epigenetic changes promote subsequent malignant progression [43]. Retinoblastoma can present with bilateral disease (involvement of both eyes) with a mean age of diagnosis of 12 months or unilateral disease (one eye involved) with a mean age of diagnosis of 24 months [10]. Due to the presence of multiple tumors, patients with bilateral or multifocal retinoblastoma can be presumed to have a germline or mosaic pathogenic *RB1* mutation. This form is considered heritable as it can be passed to future offspring [8,44] and puts affected individuals at an increased lifelong risk of developing other ocular and non-ocular tumors due to this cancer predisposition syndrome [45]. The more common presentation is with unilateral, unifocal disease; the vast majority of these children have a sporadic, somatic pathogenic *RB1* mutation which is present in the tumor only. However, as stated, 15% of these patients may still harbor a germline mutation, therefore, genetic testing on the peripheral blood is paramount [13].

The majority of pathogenic variants in the *RB1* gene either introduce a premature stop codon or cause out-of-frame exon skipping due to splice site variants. Splice donor site mutations at the CGA codon of intron 12, for example, have been observed in the setting of RB. Intronic variants, highly polymorphic microsatellites (Rb1.20, Rbi2), and minisatellites (RBD) have also been shown to abrogate gene function [46]. Over 2500 nucleotide variants have been identified in the DNA of RB patients—the majority of which are located within exons 1–25. Complex DNA rearrangements such as *RB1* gene deletions have also been observed in patients with RB and other non-ocular cancers, resulting in loss of function of the RB1 protein and cell cycle dysregulation [46,47]. A very rare and aggressive subset of unilateral, non-heritable retinoblastoma can result from highly amplified *MYCN* in the setting of normal *RB1* alleles [9].

At the cellular level, *RB1* is inherited in an autosomal recessive manner, where the inactivation of *RB1* in heterozygous state develops a normal phenotype. However, retinoblastoma phenotypically presents in an autosomal dominant manner with 90% penetrance. Penetrance of the RB phenotype is ultimately based on RB1 protein expression, which, in turn, is dependent on the type and nature of the underlying genetic mutations. A low penetrance RB phenotype is associated with in-frame variants, missense or known splice site variants, and indels in exon 1 or the promoter region [36]. Conversely, null mutations and nonsense variants typically demonstrate complete penetrance of the RB phenotype. The differential penetrance of RB is also linked to the parental origin of the pathogenic allele (i.e., the parent-of-origin effect) [36]. 

While the initiation of tumorigenesis in RB begins with a mutation in the *RB1* tumor suppressor gene, additional genetic and genomic arrangements may be required for continued tumor growth. The known mutational landscape in RB is limited, however recurrent mutations in *BCOR* and *CREBBP* have been described in a small percentage of tumors [48]. A more common mechanism for tumorigenesis in RB is somatic copy number alterations (SCNAs), including several highly recurrent alterations identified from tumor studies such as 1q, 2p, 6p, 13q and 16q. Together, these are termed RB SCNAs. The gain of 6p is particularly common in RB tumors [49] and has been associated with significantly lower rates of ocular salvage [50]. Although the specific role of these chromosomal alterations in RB development is not entirely known, it is thought that RB SCNAs influence the activity and expression of oncogenes and/or tumor suppressor genes to cause malignant progression of disease [49]. 

NGS technologies have also paved the way for investigating the role of non-coding RNA, such as miRNAs and long non-coding RNA (lncRNA), in RB development. Through NGS methods, several lncRNAs have been identified as differentially expressed in RB and potent regulators of RB progression and metastasis, including *BANCR*, *AFAP1-AS1*, *NEAT1*, *XIST*, *ANRIL*, *PlncRNA-1*, *HOTAIR*, *PANDAR*, *DANCR*, and *THOR* [51]. As sequencing technologies continue to be developed, it is critical that all genomic information from both coding and non-coding regions of the genome be considered and used to aid in diagnostic and therapeutic strategies for RB patients.

## 6. eyeGENE^®^

To facilitate research into the causes and mechanisms of inherited eye diseases and accelerate pathways to treatments and to support future research on genetic ocular pathologies such as retinoblastoma and other IRDs, a genomic medicine initiative known as the National Ophthalmic Disease Genotyping and Phenotyping Network (eyeGENE^®^) was created and launched by the National Eye Institute (NEI) in 2006 [52]. The eyeGENE^®^ network collects and manages diagnostic and clinical phenotypic data from individuals with rare inherited conditions including IRDs, in partnership with clinics and commercial and academic CLIA-approved molecular genetic testing labs performing vision research. The NEI, intramural, extramural vision research labs, and private clinics recruits affected individuals who consent to participate in research and clinical trials through eyeGENE^®^) [52,53,54]. The DNA collected by eyeGENE^®^ is sent to one of the CLIA-certified labs in the eyeGENE^®^network to perform diagnostic testing for specific disorders based on family history, retinal phenotypic presentation, and other information collected by referring eye doctors for recruited patients. The molecular genetic diagnostic reports provided by the CLIA labs to eyeGENE^®^ are then communicated back to the referring clinician. At this time, the eyeGENE^®^ has data from about 6500 participants representing over 400 clinical organizations. The Network included a patient registry, DNA biorepository and a database that allows the entry of eye health data, molecular diagnostic testing results, and linked phenotypic information [55]. All DNA samples and corresponding de-identified clinical and genetic information are available to vision researchers with an approved research project. At its inception, eyeGENE^®^ screened for 20 known disease-causing genes in nine disease categories. Currently, several hundred genes are tested through panels available through eyeGENE^®^ laboratories. As molecular testing technology evolves, so does the screening capabilities of the program. The current major focus of the program is increased data collection and continued screening for individuals already accured in the Network. For RB specifically, eyeGENE^®^ has 118 participants who have undergone genetic testing for this pediatric disease (https://eyegene.nih.gov/) [52]. The NEI eyeGENE^®^ network serves as a model for a comprehensive community partnership and as a resource for developing precision medicine strategies to treat IRDs in the future.

## 7. Modern Day Genetics and Prognostics for Retinoblastoma

Despite the increased focus on molecular diagnostics for IRDs, for retinoblastoma this is critically limited by the inability to directly biopsy retinoblastoma tumors due to concern for extraocular spread. While germline *RB1* mutations can be isolated from the peripheral blood and used to stratify the risk of second ocular and non-ocular tumors for patients with heritable disease, this has little to no impact on the diagnosis of retinoblastoma, nor on the prognostication for globe salvage. There is an active clinical trial evaluating whether certain germline *RB1* mutations affect the risk of secondary non-ocular tumors (ClinicalTrials.gov NCT00342797), but this is not specific to ocular prognostication.

Various genetic and genomic aberrations have been described from studies on tumor tissue in enucleated eyes with retinoblastoma [48,56]. However, as therapeutic efficiency increases, the rate of enucleation is decreasing and thus, tumor tissue is less available. Advances in liquid biopsy in cancer diagnosis and prognosis have recently been applied to retinoblastoma using the aqueous humor (AH) as the biofluid of choice. In 2017, Berry et al. [57] demonstrated that tumor-derived cell-free DNA could be identified in this fluid due to a copy number profile of chromosomal gains and losses that was concordant with the DNA from the tumor (obtained from the eye after enucleation). For the first time, this opened the door to obtaining tumor-specific molecular biomarkers in vivo*—*either at diagnosis or during globe-salvaging treatment. The authors went on to analyze 63 AH samples from 29 eyes of 26 patients; 13 eyes were enucleated and 16 were saved [50]. This approach allowed for a direct comparison of the genomic profiles between the eyes that responded to therapy and were saved versus those that subsequently required enucleation. The authors found that gain of 6p was the most commonly seen SCNA in the AH, with a statistically significant difference for this biomarker in enucleated eyes (77%) versus salvaged eyes (25%) (*p* = 0.0092). Notably, if gain of 6p was present in the AH of an eye with retinoblastoma, it was associated with 10x increased odds of that eye requiring enucleation (OR 10, 95% CI 1.8–55.6) [50]. This was the first study to suggest that genetic and genomic evaluation of the AH could be used for prognostication of clinical outcomes, and to describe gain of 6p in the AH as a biomarker (Figure 1).

The Berry et al. [50] study also demonstrated several parameters that have the potential to be critically important to the future of molecular diagnostics and prognostics in retinoblastoma. First, rare focal genomic alterations, such as *MYCN* amplification, can be detected, which can be additionally impactful for prognosis of this disease. Secondly, evaluation of longitudinal AH samples can be used to demonstrate tumor response with a decrease in the tumor-derived cfDNA fraction in these eyes and a concomitant normalization of the genomic profiles. Alternatively, relapse can also be predicted by an increase in tumor fraction and the amplitude of SCNAs. Finally, Gerrish et al. recently demonstrated that the DNA extracted from the AH can be used to isolate the tumor specific *RB1* mutation(s) in the absence of tumor tissue [58]. This critical advance allows for the identification of both somatic *RB1* mutations, which can aid in tumor-directed mutational analysis of the peripheral blood. This is especially useful for detecting low-level (<5%) mosaicism. It should be noted that this modality is currently not in clinical use and is done only in a research setting.

Now that we have safe access to a source of tumor-derived DNA and other biomarkers for retinoblastoma in the AH, the recent advances in molecular testing for IRDs and in liquid biopsy platforms in other cancers can be applied to this cancer. Just as the cloning of the *RB1* gene spurred a change in our understanding of cancer genetics and tumor suppressor genes, so these recent developments have led to an exciting new field with the potential application of specific diagnostics and prognostics for retinoblastoma (Figure 1). Hopefully, this will be a critical step toward precision therapies for retinoblastoma based on an improved understanding of the mechanisms for treatment response, resistance and recurrence in this inherited retinal cancer.

## 8. Patents

Provisional Patent Application filed: Aqueous humor cell free DNA for diagnostic and prognostic evaluation of Ophthalmic Disease (Jesse L. Berry).

## Figures and Tables

**Figure 1 genes-10-01032-f001:**
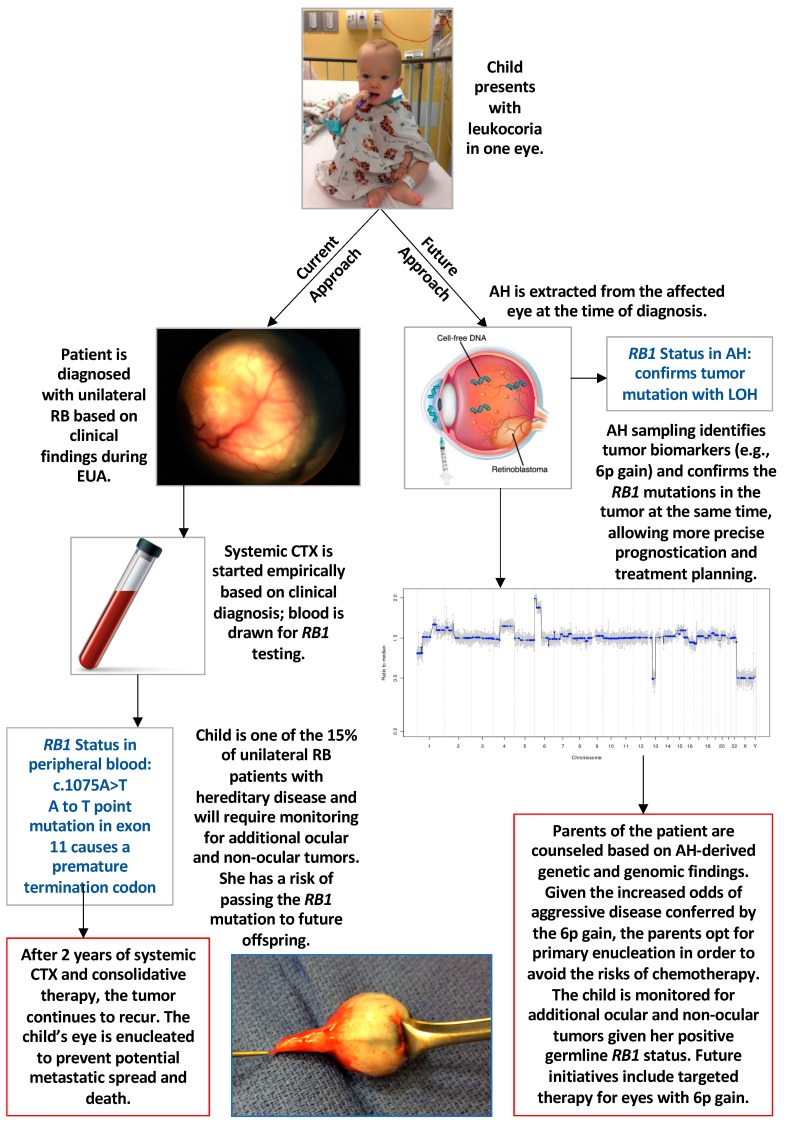
Current and future approaches to retinoblastoma (RB) diagnosis and management. Abbreviations: AH, aqueous humor; CTX, chemotherapy; EUA, examination under anesthesia; LOH, loss of heterozygosity; RB, retinoblastoma.

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
