# Peer review of "Next-Generation Technologies and Strategies for the Management of Retinoblastoma"

_genes, 2019, doi:10.3390/genes10121032_

Round 1
Reviewer 1 Report
Gudiseva, H. V. et al. describe the current state with regards to the diagnosis of retinoblastoma (RB), the treatments, and some recent advances that could potentially change the current practice for the diagnosis and management of RB. The review is clearly written and is important for the field of RB research and therapy. I only have one request for the authors, which is to enlarge the only figure in this review in order to make it more clear for the readers.
Author Response
Reviewer 1
Gudiseva, H. V. et al. describe the current state with regards to the diagnosis of retinoblastoma (RB), the treatments, and some recent advances that could potentially change the current practice for the diagnosis and management of RB. The review is clearly written and is important for the field of RB research and therapy. I only have one request for the authors, which is to enlarge the only figure in this review in order to make it more clear for the readers.
Response: We thank the reviewer for this positive feedback. We have modified the figure so that it can be printed as a full page and made it clearer for the readers.
Reviewer 2 Report
Overview
This is an informative review that details an exciting new diagnostic approach to retinoblastoma.
Key Points
This review is poorly referenced. Firstly, the references are too few. Secondly, 16 of the 31 references are 5 years old or more. This results in a largely unsupported commentary of current retinoblastoma research. This review would be radically improved if the text could be supported by additional evidence.
Specifics
Introduction
L37: Edit- Over 270 genes now
L39: There are more appropriate and recent studies that include RB1 detection by whole-exome sequencing.
L41: Reference needed for claim of "prototypical IRD".
L44: That stat (40%) is nearly 30 years old now.
Retinoblastoma
L52: Repeat of "prototypical" claim. Again it is unsupported.
L62: No clear reference for incidence stat.
L63: If using ref8 to support this statement, the reference is to a paper that specifically states "incidence is constant worldwide".
L69: Ref is 15 years old and predates any mainstream utility of NGS etc., to diagnose genetic disorders, therefore is unlikely to still be accurate.
L75: This statement is undermined by L71-72.
Retinoblastoma Therapy
L81: Define high-risk clinical features and reference appropriately.
L83: Rephrase "no view due to".
L85: "Must include" implies there is a standard of care that involves these treatments. This articles are published annually for some countries. An appropriate reference should be available.
L93: This statement is contradicted by L71-72.
L95: Elaborate on the nature of the screening (eg. imaging, genotyping, etc)
Molecular Testing in Retinoblastoma
L101: This statement requires multiple recent publications to support it.
L106: The only reference in this paragraph is from 1993. That is not sufficient.
L120: Ref15 is a review that simply notes the findings of Ref13 and Ref14, please remove.
Retinoblastoma Genetics and Genomics
L137: This paragraph is unclear and would benefit from clarifications regarding germline versus somatic RB development (and additional references). Ref15 is a review, please use original publications. Ref5 is nearly 30 years old.
L147: This statement needs a reference. Also, this entire paragraph appears to rely heavily on a dated reference (#21-2011). This means that these findings are not suited to phrases like "to date" (line 151).
L159: This statement contradicts the possibility of a dominant germline inheritance pattern as suggested in Ref2. This statement also requires a reference.
L170: The (insert: known) mutational landscape...
eyeGene
L209: Typo- eveolves
L211: Why isn't this cohort used to generate more current statistics regarding incidence of somatic versus germline pathogenic variants?
Modern Day Genetics ...
General: This section very heavily focuses on aqueous humour based methodologies, are there no other techniques utilised or in development (see clinicaltrials.gov). More importantly, this section appears very biased towards this method. The limitations of this approach should also be discussed as this is a review.
L216: This paragraph would benefit from being rephrased as it currently flows quite poorly.
L219: Where stats are quoted, they should be supported by references. There is no reference in this paragraph currently.
Author Response
"Please see the attachment"
